# Seasonal and Geographic Variation in Alkaloid Content of Kratom (*Mitragyna speciosa* (Korth.) Havil.) from Thailand

**DOI:** 10.3390/plants12040949

**Published:** 2023-02-19

**Authors:** Narumon Sengnon, Phanita Vonghirundecha, Wiraphon Chaichan, Thaweesak Juengwatanatrakul, Jumpen Onthong, Pongmanat Kitprasong, Somchai Sriwiriyajan, Somsmorn Chittrakarn, Supattra Limsuwanchote, Juraithip Wungsintaweekul

**Affiliations:** 1Department of Pharmacognosy and Pharmaceutical Botany, Faculty of Pharmaceutical Sciences, Prince of Songkla University, Hat Yai Campus, Songkhla 90112, Thailand; 2Narcotic Crops Survey and Monitoring Institute, Office of the Narcotics Control Board, City Hall, Muang, Chiang Mai 50303, Thailand; 3Faculty of Pharmaceutical Sciences, Ubon Ratchathani University, Warinchamrab District, Ubon Ratchathani 34190, Thailand; 4Agricultural Innovation and Management Division, Faculty of Natural Resources, Prince of Songkla University, Hat Yai Campus, Songkhla 90110, Thailand; 5Division of Health and Applied Sciences, Faculty of Science, Prince of Songkla University, Hat Yai Campus, Songkhla 90110, Thailand

**Keywords:** kratom, *Mitragyna speciosa* (Korth.) Havil., mitragynine, speciogynine, paynantheine, seasonal variation, geographical origin, Thailand

## Abstract

The objective of this study was to obtain data on the distribution of alkaloids in kratom plants grown in Thailand. Two collections were performed, covering the southern, central, and northern regions of Thailand and different seasons. The contents of alkaloids, including mitragynine (MG), paynantheine (PAY), and speciogynine (SG), were determined using the validated HPLC method. The 134 samples in the first collection were collected from Nam Phu subdistrict, Ban Na San, Surat Thani, Thailand, during June and October 2019 and January 2020. The maximum mitragynine content was 4.94% *w*/*w* in June (late summer), and the minimum content was 0.74% *w*/*w* in October (rainy season). To expand the study area after kratom decriminalization, 611 samples were collected in June–August 2021, October–December 2021, and January–April 2022. The accumulation of MG ranged from 0.35 to 3.46% *w*/*w*, 0.31 to 2.54% *w*/*w*, and 0.48 to 2.81% *w*/*w*, respectively. The meteorological data supported the climate’s effect on alkaloid production. Soil analysis revealed the importance of Ca and Mg in promoting alkaloid production. Geographical locations played a role in the variation of MG in kratom leaves, but did not affect the color of leaf veins. In conclusion, the present study suggested that the alkaloid content in kratom diverges based on seasonal and geographical origin.

## 1. Introduction

Kratom, or *Mitragyna speciosa* (Korth.) Havil., belongs to the family Rubiaceae. Thailand is home to five *Mitragyna* spp., including *M. speciosa* (Korth.) Havil., *M. diversifolia* (Wall. Ex G. Don) Havil., *M. hirsuta* Havil., *M. rotundifolia* (Roxb.) Kuntze, and *M. parvifolia* Korth. [1,2]. Kratom is an evergreen tree and can grow up to 30 m tall in wet and humid conditions, with medium to full sun exposure. The leaves of Thai kratom have at least two types according to the leaf vein: red-veined and green-veined, with some having a serrated apex [2,3,4,5]. The most abundant active pharmaceutical ingredients in red- and greed-veined and apex-serrated leaves are still under discussion. However, the natives of southern Thailand believe that the red-veined leaves are useful for medicinal purposes and that the green-veined leaves are energy-boosting [6]. Nevertheless, the genetics of these types of the leaves have not been discriminated by molecular techniques to date [7,8].

Kratom is distributed widely in Southeast Asia, including Thailand, Malaysia, Indonesia, Borneo, and the Philippines [2,9]. In Thailand, the natural distribution of kratom is found mainly in the south, and it is abundant along riverbanks and in national forests. The existing kratom in other regions of Thailand is primarily distributed by humans or in cultivation. The laborers in rubber fields chew kratom leaves before work as a stimulant and to relieve muscle pain [3]. Kratom is traditionally used in Thai medicine, as documented in Thai Materia Medica. Prasakratom, for instance, is largely composed of kratom leaves and is prescribed for diarrhea. In folklore medicine, kratom has been used primarily to treat pain and diarrhea and to lower blood sugar [6]. 

More than 25 alkaloids are reported in kratom, mainly including mitragynine, representing about 66% *w*/*w* of the crude alkaloid extract prepared from Thai kratom, while it represents about 12% *w*/*w* of that prepared from Malaysian kratom [10]. Mitragynine is a major alkaloid found in the leaf which is biosynthesized by means of the Mannich reaction between tryptamine and secologanin, catalyzed by strictosidine synthase [11,12]. Tryptamine from the shikimate pathway has been shown to limit the production of mitragynine in plants [13]. The alkaloid contents in kratom leaves (based on alkaloid extract) range from 66% for mitragynine (MG) to 9% for paynantheine (PAY), 7% for speciogynine (SG), 2% for 7-hydroxymitragynine, and 1% for speciociliatine [14] (Figure 1). Kratom leaves also contain tannins, flavonoids, triterpenoids, and other phenolic compounds [15,16]. Examination of kratom plants revealed that mitragynine is abundantly accumulated in the leaves, less so in the stem, and not present in the roots [13].

Mitragynine acts as an analgesic via binding to opioid receptors. Mitragynine is changed by hepatic cytochrome P450 into the active metabolite 7-hydroxymitragynine and passes through the blood–brain barrier into the central nervous system [17,18,19]. Kratom extract was shown to have an antidiarrheal effect in castor oil-induced diarrhea in rats [20]. A study on muscle cells revealed that kratom extracts induce the reuptake of glucose in the cell by inducing the expression of the glucose transporter [21]. However, there is uncertainty regarding the mechanism of reducing blood sugar, which requires more clinical research. 

Kratom was illegal in Thailand for many years [22]. In August 2021, the Narcotic Act was revised, with kratom being removed from the list of level 5 narcotics during the legislative reform of drug policies. After this, Thai researchers were able to investigate kratom without permission from the Thai FDA. On 26 August 2022, the Thai government gazette announced the ‘Kratom Act,’ controlling the trade of kratom internationally. Currently, the Thai government urges extensive research into and development of kratom for medical purposes. Therefore, investigation of Thai kratom, regarding its distribution and chemical profile, is of great importance. 

During the period of 2018–2020, the Narcotic Crop Survey and Monitoring Institute (NCS), the Office of the Narcotics Control Board (ONCB), began to develop an application to search for kratom trees in Thailand, especially in the south. The authority gave permission to grow kratom in Nam Phu subdistrict, Ban Na San, Surat Thani. NCS registered 1578 kratom trees, and the survey application built a QR code which could store information including location (latitude–longitude), size (height, girth), age, and even the owner of the kratom plant. The initiation of the legislative reform was begun by means of reinforcement from the government, and transforming kratom from a narcotic into a medicinal plant was based on the knowledge of the relevant pharmacologically of the active compounds in kratom plants, along with their traditional usage. It can be noted that the decriminalization of the utilization of kratom does not cover its combination with psychoactive substances or drugs, which are still illegal. 

To support the survey data on the distribution of kratom and accumulating alkaloids, the present study aimed to collect kratom leaves from across Thailand. Two stages of collection were performed before and after decriminalization, covering the southern, central, and northern regions of Thailand and different seasons. Most kratom plants in this study were naturally grown in households (not as a commercial crop). The high-, medium- and low-alkaloid-producing kratoms were grouped, and we tried to explain the factors by analyzing soil nutrients and the climate. The macroscopic of kratom leaves is illustrated and histolocalization of alkaloid is determined. Our results will support the knowledge about the qualities of Thai kratom, which is essential for pharmaceutical applications.

## 2. Results

### 2.1. General Appearances and Macroscopic Characteristics

We first investigated the distribution of alkaloids in kratom leaves in three different periods, including June 2019, October 2019, and January 2020. The permitted area for kratom cultivation according to the ONCB, Thailand, is Nam Phu subdistrict, Ban Na San, Surat Thani. Figure 2 shows the location of the subdistrict. In total, 134 samples were randomly collected from the areas M1–M6 (Set 1, Appendix A). The populations of green-veined and the red-veined varieties were 75/134 (56%) and 59/134 (44%), respectively. In interviews, the owners stated that the kratom plants originated from seedlings, grafting, or cuttings. The trunk diameters (girth) varied from 10 to 95 cm (Appendix A). The young woody shoots bore 10–12 leaves arranged in opposite and decussate pairs. Each pair of leaves was accompanied by two interpetiolar stipules, which initially protected the apical bud. The leaves were ovate-acuminate in shape with 12–17 pairs of veins (Figure 3). The leaves appeared dark green to green and leafy or glossy and could grow to over 8–20 cm (3.2–7.9 inches) long and 6–12 cm (2.4–4.7 inches) wide when fully open. The color of the stipule was green or red. Weighing fresh leaves found that the size ranged from 1 to 4 g, or an average of 2.1 ± 0.5 g per leaf. The drying index was about 3.4. After drying at 50 °C for 24 h, the average dry weight was about 0.6 ± 0.1 g per leaf. 

### 2.2. Histolocalization of Alkaloid 

Upon histochemical analysis on the transverse section of leaves and petioles, we found an accumulation of alkaloids in the leaf palisade and spongy parenchyma after dyeing with a modified Dragendorff TS2 solution. In addition, the idioblast containing alkaloids also appeared in the cortex and pith of the petiole (Figure 4).

### 2.3. Establishment and Validation of HPLC Analysis

Mitragynine, paynantheine and speciogynine were isolated in-house from red-veined kratom leaves. Appendix A summarizes the isolation protocol of alkaloids, and describes the alkaloid characteristics. The HPLC chromatogram was performed through isocratic elution with an acetate buffer and acetonitrile. The retention times of speciogynine, paynantheine, and mitragynine were 9.4, 10.3, and 14.1 min, respectively (Figure 5). According to the ICH guidelines, the linearity, precision, accuracy, limit of detection (LOD), and limit of quantification (LOQ) are shown in Table 1.

### 2.4. Distribution of Alkaloids in Kratom Grown in Nam Phu Subdistrict

Kratom samples were collected (Appendix A) and prepared for HPLC analysis. Contents of MG, PAY, and SG were calculated, and the summation of three alkaloids was expressed as the total alkaloids. Figure 6a indicates that the samples’ distribution of alkaloids varies from maximum to minimum in each period. Specifically, regarding the MG content, the maximum yield of MG of 4.94 ± 0.43% *w*/*w* was observed for the sample M6-3 in June 2019. The minimum value of MG of 0.74 ± 0.04% *w*/*w* was obtained from the sample M3-7 in October 2019. The plot of MG content suggests that the median MG content was 2.16% *w*/*w*.

The amount of MG, PAY, and SG significantly differed based on the collection period. The population of kratom was distributed with a high MG content mainly in June 2019, and the lowest MG content was observed in October 2019. The accumulation of MG, PAY, and SG was similar in each group. Figure 6b shows the accumulation of alkaloids was high in June 2019, followed by January 2020, and October 2019. To observe the ability of alkaloid production by kratom, the MG:PAY, MG:SG and PAY:SG ratios were plotted against time of harvesting. Figure 6c shows that PAY significantly affected the total alkaloid content between June, October, and January (*p* < 0.01). Multiple comparisons of MG, PAY, and SG contents and the color of the leaf veins (Figure 6d) suggested that the production of the three alkaloids in each group was consistent in red- and green-veined kratom plants in each period.

### 2.5. Distribution of Alkaloids in Kratom across Thailand

Kratom samples, obtained in June–August 2021, October–December 2021, and January–April 2022, were collected from the same kratom tree to understand the seasonal change in alkaloid production. All samples were harvested from 13 provinces, 14 districts, and 19 subdistricts covering the north, center, and south (west coast and east coast) of Thailand (Appendix A). Samples comprised 57% green-veined, 29% red-veined, 5% serrated red-veined, 3% serrated green-veined, and 6% unspecified leaf-veined kratom plants. Figure 7 summarizes the distribution of MG, PAY, and SG in the 611 kratom samples. The highest MG level in a sample was about 3.46% *w*/*w* in S-TMN5, collected in June–August 2021, and the lowest MG level in a sample was about 0.39% in S-PN5, harvested in October–December 2021. It can be noted that kratom sample No. R-LL6 has a unique pattern of alkaloid production, as it contains MG 0.90% *w*/*w*, PAY 0.18% *w*/*w*, and SG 2.40% *w*/*w*. This was a sample from M3, Laun Nuea subdistrict, Laun district, Ranong province on the west coast of southern Thailand.

Samples from kratom harvested during June–August 2021 contained more MG, PAY, and SG than those harvested in other periods. In addition, kratom collected in October–December 2021 contained alkaloids at the same levels compared to samples collected in January–April 2022 (Figure 8a). In a similar pattern, the plots of the MG:PAY, MG:SG, and PAY:SG ratios in Figure 8b reveal that samples collected in June–August 2021 produced MG or PAY at higher levels than another periods of collection (Figure 8b). Analysis of leaf vein color and alkaloid production indicated no difference between each collection group. Notably, in samples collected in June–August 2021, MG and PAY were distributed in the green-veined population rather than the red-veined population (Figure 8c).

### 2.6. Meteorological Data and Geographic Variation in Alkaloid Content 

Data on average rain precipitation and temperature for June 2021–May 2022 were retrieved from the Thai Meteorological Department, Thailand, to understand their effects on alkaloid production. The weather stations used in this study were at Petchabun, Bangkok (Don Muang), Ranong, and Surat Thani, denoted as north (N), central (C), west coast of south (WC), and east coast of south (EC), respectively. Figure 9 summarizes the average precipitation in mm and the average temperature from June 2021 to May 2022.

The average precipitation differed at each weather station. From June 2021 to May 2022, abundant rainfall was observed in the WC region, with an average value of 497 mm, whereas 160 mm was observed for EC, 123 mm was observed for central, and 93 mm was observed for the north. The fluctuation of precipitation found in the WC region began with ca. 814 mm in June–October 2021 and lower rainfall of about 124 mm during January–April 2022. The minimum rainfall was observed in January 2022, and the maximum was observed in August 2021. In contrast, the rain came later in the EC region, starting in October–December 2021. The highest rainfall in the EC region was observed in November 2021. The upper part of Thailand, namely the central and northern regions, has a lower amount of rain when compared to southern Thailand. The average temperature from June 2021 to May 2022 was about 28.8 ± 0.5 °C for the whole country. The weather station at Petchabun recorded a temperature range of 23.5–31.7 °C in December 2021 and June 2021, respectively. Every part of Thailand had the lowest temperature in December 2021 and the highest temperature in April 2022. It is notable that southern Thailand, both in the WC and EC regions, has less temperature variability. 

Data for total alkaloid content were re-arranged according to the locations. The north set includes samples from Chiang Mai, Lampoon, and Petchabun. The central set includes samples from Pathum Thani and Nonthaburi. Most data regarding kratom were from the south, divided into the WC region (Phangnga, Krabi, Trang, and Ranong) and EC region (Prachuap Khiri Khan, Chumphon, Surat Thani, and Nakhon Si Thammarat). During June–August 2021, the lowest amount of alkaloid was found in the WC region. On the other hand, there was no difference between the N, C, and EC regions (Figure 10). The results of total alkaloid contents in October–December 2021 and January–April 2022 in kratom tended to be lower than in June–August 2021. The distribution of alkaloids was observed in the population from the EC region in every season. 

### 2.7. Effect of Soil Nutrients on Alkaloids Contents 

We aimed to collect 20 samples from different locations based on the MG contents obtained from samples in June–August 2021. The samples were divided into three groups according to MG contents of <1%, 1–1.5%, and >1.5% *w*/*w*. In parallel, we collected kratom leaves and soil from the same site in January–April 2022. The production of MG, PAY, and SG in kratom exhibited different profiles (Figure 11). In the HPLC chromatograms, the area of the peak corresponds to the calculated amount of a substance, and Figure 11 indicates the divergent chemical profiles in kratom samples. The ratio of MG:PAY:SG was typically 7:1:1, except for R3-LL6, a high-SG-containing kratom, for which the ratio of MG:PAY:SG was 0.5:0.1:1. Kratom samples were determined for alkaloid contents, soil nutrients, as well as analysis of soil type and water holding capacity (Table 2 and Table 3). 

Soil analysis, including the determination of the pH, electric conductivity, macronutrients (N, P, K, Ca, Mg), micronutrients (Fe, Mn, Zn, Cu), organic matter, soil type, and water holding capacity, was conducted (Table 2 and Table 3). The soil pH varied from 4.89 to 6.48 (average 5.57 ± 0.39). The electric conductivity (EC) ranged from 0.014 to 0.682 dS/m. The soil organic matter varied from low to moderate (2.5–30.7 g/kg). The soil types included clay, clay loam, sandy clay loam, silty clay, and silty clay loam. Sandy clay loam had less water-holding capacity when compared to clay and silty clay. The color of the soil was from light brown to dark brown. Principle component analysis (PCA) of all parameters, shown in Figure 12, reveals that MG and PAY contents were related to total alkaloids, depended on pH, EC, and macronutrients. In contrast, micronutrients and organic matter had less influence on alkaloid production. Kratom plants containing high SG levels had a lower production of MG and PAY.

Multivariate linear regression (1 dependent, n-dependent) suggested that MG correlates with total alkaloid content and PAY (*p* < 0.001). Macronutrients such as Ca and Mg have huge variation and affect MG content with a p-value of 0.223 and 0.182, respectively. Univariate analysis (ANOVA; Tukey’s pairwise) found that the amount of Ca significantly affects MG, PAY, SG, and total alkaloid production. Whisker box plots of soil type and soil nutrients indicated the variability values of each parameter (Figure 13a) among 20 samples. The soil nutrients in different locations varied greatly. This showed that soils with higher P, Ca, Mg, Fe, and Mn tended to give a higher alkaloid content. Classical clustering and applied Euclidean distance as a similarity index classify kratom samples into three groups, with low, medium, and high alkaloid production (Figure 13b).

## 3. Discussion

For almost 80 years, Thai kratom was illegal according to its status as a level 5 narcotic plant. Meanwhile, several researchers around the world have reported the pharmacological activities and plausible mechanisms of this plant in in vitro and in vivo models [4,9]. The observational studies suggested that consumers applied kratom for pain relief, sugar blood control, and the treatment of drug withdrawal symptoms. Acceptance of its benefits rather than its psychoactive effects is the main reason for the government considering legislative reform of its drug policy [22]. After five years of preparation, on 24 August 2021, kratom was removed from the narcotic list, and it is now under the control of the ‘Kratom Act 2022′. Currently, many campaigns are ongoing to promote kratom to as a new economic crop in Thailand. Therefore, people are focusing on kratom in many aspects, including foods, dietary supplements, herbal products and medicines. Kratom should be used carefully, concerning its relevance, effectiveness and safety. We cannot deny that kratom has a psychoactive effect and care needs to be taken when combining it with other drugs or psychoactive substances. However, Thai people realize this and have a good attitude toward kratom [23]. The southern natives usually chew kratom leaves before going out and working in the field, with the purpose of increasing energy and preventing muscle pain [24]. Furthermore, kratom is an ingredient combined with other herbs in Thai traditional medicine for gastro-intestinal ailments [9,25]. Although kratom is now legal in Thailand, but in many countries kratom is still banned, such as in Myanmar, New Zealand, Australia, Korea, etc. [26]. Kratom leaves contain a wide variety of substances, such as alkaloids, tannins, flavonoids, and triterpenoids. More than 25 alkaloids have been isolated from kratom leaf, and among them the prominent alkaloid is mitragynine (MG), followed by paynantheine (PAY), speciogynine (SG), 7-hydroxymitragynine (7-OH-MG), speciociliatine and others found at less than 1%, including mitraphylline, rhynchophylline, and ajmalicine. The total alkaloid content of kratom leaves varies from 0.5 to 1.5% *w*/*w* [14,16].

Kratom is a native plant in Myanmar, Thailand, Malaysia, and Indonesia. Thai kratom is found abundantly in southern Thailand [2]. Red-veined and green-veined kratom plants are plentiful in the area. A summary of alkaloid distribution in Thailand is depicted in Figure 14. Investigation of 745 samples, collected in the years 2019–2022, demonstrated that the MG content varies from 0.39% to 3.46%. In the first part of the investigation, kratom samples were randomly collected from the blocks M1–M6 in Nam Phu subdistrict. They were harvested in three periods, namely June and October 2019, and January 2020. Figure 6a clearly shows that kratom produced high levels of alkaloids in June and produced lower levels of alkaloids in October. The hypothesis of seasonal variation in alkaloid production was raised. After kratom decriminalization, we could obtain samples from other parts of Thailand. To reduce the variability of alkaloid production, we collected the samples from the same plant at different times. Ten samples from each location were harvested at the selection period (Appendix A). The HPLC analysis data suggested that kratom accumulated high MG and total alkaloid contents in June–August 2021 significantly (Figure 7 and Figure 8). From this evidence, it can be concluded that alkaloid production, i.e., MG, PAY, and SG, varies dependent upon seasonal variation.

There are three seasons in Thailand, i.e., summer, rainy, and winter. Since the weather is influenced by northeast and southwest monsoons, the temperature does not dramatically change throughout the year. The monsoons create a hot season (March–April), wet season (October–December), and cool season (January–February). In southern Thailand, in particular, influences from the northeast monsoon in the Andaman Sea and from the southwest monsoon in the Gulf of Thailand create two seasons: the hot and wet seasons. The rainy season in the WC is during July–November, while in the EC region it comes later in October–December. Our results from the samples collected in June–August 2021 show that the lowest amount of MG production observed was in the WC, while the highest amount of MG production was observed in the EC (Figure 10). Precipitation played a vital role in the total alkaloid content, which resulted in the production of lower levels of alkaloids in kratom plants in the rainy season. The tide washed out the soil nutrients, and probably affected the production of alkaloids [27]. This evidence was also found in the alkaloid analysis of the samples collected across the country. The heavy rain comes earlier in the year in the WC of southern Thailand, from July until October. As a result, the seasonal change influenced the alkaloid production. Another reason that causes the lower alkaloid production in the southern area was the influence of temperature, altering alkaloid production during December 2021–January 2022. Kratom leaves appeared green and large. The extent of medium to full sunlight exposure was a crucial parameter enhancing alkaloid production [28]. In the northern region, the cool and dry season is from December to January, during which kratom produces a lower alkaloid content. The present study revealed that the kratom leaves produced and accumulated the total alkaloid content, MG in particular, in the late summer, between the hot and rainy season. 

The correlation between the location of a kratom tree and the alkaloid content indicated that the southern part of the EC region is a good location for kratom growing, followed by the central region, as shown in Figure 10. In contrast, kratom plants in the north and the south (WC) need intensive care regarding soil nutrition and the environment. Previously, Leksungnoen et al. (2022) reported the variation of MG content in kratom plants grown in different parts of Thailand. The MG contents, collected from September to December 2021, were 1.8% in the south, 2.2% in the central region, 1.2% in the north, and 1.1% in the northeast [29]. Notably, the samples from the south were from Ranong, a province in the WC region. In our study, focusing on ten samples from Ranong province, nine samples showed a similar pattern of alkaloid production, except for R-LL6, which produced SG as a principal component. The average MG contents were 1.8% in samples collected in June–August 2021, 1.3% in October–December 2021, and 1.6% in January–April 2022. Overall, the distribution of alkaloids in kratom grown in Thailand is dependent upon geographical origin. The EC region of southern Thailand would be a good location for producing high-MG kratom. The present study suggests cultivating kratom in Chumphon, Surat Thani, Nakhon Si Thammarat, and Prachuap Khiri khan.

The MG content in kratom leaves was reported from countries in Southeast Asia. Previously, we reported that the MG content in plants from Surat Thani (Panom district) varied from 1.6 to 5.5%, whereas in Nakhon Si Thammarat it was about 1.3% and in Satun was 0.8% [30]. Chear et al. (2021) observed MG contents ranging from 0.9 to 1.8% (Penang, Malaysia) and from 1.1 to 1.7% (Kedah, Malaysia) [31]. The MG content of Indonesian (Kalimantan) kratom ranged from 0.37% to 1.70% [32]. Along the Malay Peninsula, kratom trees are distributed in the coastline region, especially in the upper part of southern (WC and EC) Thailand, Penang and Kedah in northern Malaysia, and Borneo Island and east and west Kalimantan in Indonesia. The kratom trees of these regions produce a considerable amount of MG. Monsoon weather suits kratom growth, and an abundance of sunlight promotes alkaloid production. Kratom in Southeast Asia, therefore, is quite attractive and a good source for further pharmaceutical usage. In contrast, kratom plants grown in USA (greenhouse, Florida, USA) contained an MG amount of 0.018% per leaf dry mass [27,28]. Thus, growing kratom trees requires a tropical environment that is dependent upon geographical origin. 

The question of the difference in alkaloid production in red-veined and green-veined kratom was answered in our study. Figure 6d and Figure 8c indicate that the alkaloid production is similar in both types of leaf, without significant differences. The green-veined leaves have a larger population than red-veined leaves. The color of leaf veins might be caused by the soil type and nutrients, and no genetic discriminant can be observed between the two types of leaves. There was a report showing that the difference in the color of leaf veins probably depends on the development stage of the leaf, not different varieties [2]. The histolocalization of alkaloids was estimated by staining with modified Dragendorff TS2. In both the leaf lamina and petioles, parenchyma cells that took up the stain were present in palisade and spongy parenchyma cells, the cortex and pith. In the same manner, localization of alkaloids in *Cinchona ledgeriana* was shown in the idioblasts in the leaf hypodermis and leaf palisade [33]. 

The samples were then collected from various locations, covering low-, medium-, and high-MG content kratom plants to understand the effect of soil nutrients on alkaloid production. The HPLC chromatograms in Figure 11 indicate the variability of alkaloid production in the samples. Except for sample code R-LL6 (Laun district, Ranong province), the ratio of MG:PAY:SG in 19 samples was approximately 7:1:1, accounting for 78%, 11%, and 11% of the content, respectively. Malaysian kratom had a ratio of MG:PAY:SG of 10:1:1 as per a previous report [31]. In addition, there are unusual alkaloids reported from Malaysian kratom, such as mitragynaline, corynantheidaline, mitragynalic acid, and corynantheidalinic acid [10]. Interestingly, kratom grown in US has a MG:PAY:SG of roughly 0.2:0.2:1 [27]. The growth environment could cause this dissimilar alkaloid profile. From our investigation, sample code R-LL6, collected from Laun district, Ranong, contained SG as a prominent alkaloid rather than MG. This rare kratom, found to have a contradictory alkaloid profile, showed a ratio of MG:PAY:SG of 0.5: 0.1: 1. Recently, at least two subpopulations of Thai kratom were examined based on a chromosome genome assembly, which were collected from central and southern Thailand [34]. The authors concluded that kratom growth is dependent upon geographical origin. Our findings confirm that the alkaloid profiles in kratom varies according to many parameters, such as seasonal change, soil nutrients, and geographical origin. Certainly, these factors have an effect on alkaloid biosynthesis, specifically on the expression of biosynthetic genes. The presence of alkaloids in kratom leaves in different patterns may cause the alteration of pharmaceutical properties and herb–drug interactions. The crude samples, obtained from venders in the US, showed different profiles of alkaloids, such as MG and speciofoline. Their finding summarized the safety and toxicity concerns that we should be aware of regarding the presence of alkaloids, such as MG and speciofoline [35]. The distribution of alkaloids in kratom of the region of Southeast Asia especially in Thailand depended on the seasonal change. Generally, kratom can propagate from seeds, however, cutting and rooting was popular for fast propagation. Beyond that, we can find kratom, obtained from grafting—either grafting on *M. speciosa* or *M. diversifolia*. Therefore, the kratom obtained from different origin may alter the alkaloid profile and subsequently the therapeutic value and safety profile. 

## 4. Materials and Methods

### 4.1. Plant Materials

Kratom leaves were kindly provided by the Narcotic Crop Survey and Monitoring Institute (NCS), the Office of Narcotic Control Board (ONCB). The protocol for collection was established by the research team, ONCB, and the communities (Appendix A). The samples were collected in different seasons in Thailand. Only healthy leaves were gathered, excluding those with mold, cast skins, lace-like injury, shredding or any signs of deterioration. Leaves were air-dried at room temperature for three days, sealed, and sent to the laboratory. After receiving the samples, kratom leaves were inspected, and dried in a hot air oven at 50 °C for 24 h, and stored in a dried place until use.

The first set (Set 1) was collected during June 2019–January 2020 from six villages (M1–M6) in Nam Phu subdistrict, Ban Na San district, Surat Thani, Thailand (GPS:8°45′ 15.71″ N, 99°17′ 7.71″ E). A total of 134 samples were collected in three different periods: 1. June 2019; 2. October 2019; 3. January 2020. A list of the samples is shown in Appendix A. The collection was performed under the Thai FDA permission No. 11/2562 issued on 1 January 2019 (ONCB, Ministry of Justice). For processing and analyzing kratom, the Faculty of Pharmaceutical Sciences, Prince of Songkla University, held the Thai FDA permission No. 9/2562 and 17/2562, issued on 14 January 2019. The information relating to kratom trees was recorded by interviewing the owners, including the origin of kratom plant, its girth, approximate height, estimated age, and the color of the veins of its leaves (Appendix A). The herbarium specimens were prepared from red-veined kratom (N5/001, N6/004, N6/005, N6/006) and green-veined kratom (N1/001) (see Appendix A). The voucher specimens were identified by Associate Prof. Dr. Kittichate Sridith and deposited at the PSU Herbarium, Department of Biology, Faculty of Science, Prince of Songkla University, Songkhla, Thailand. After decriminalization and reform of the Narcotic Act, the second set (Set 2) was expanded by collecting from 19 subdistricts in 13 provinces: 9 in the south; 2 in the central region; and 2 in the north. Again, the collections were carried out following the protocol as described above. A total of 611 samples were collected in a similar manner: 1. June–August 2021; 2. October–December 2021; 3. January–April 2022. The list of samples is summarized in Appendix A.

### 4.2. Chemicals and Instrumentations

Solvents were of analytical and commercial grade, and were distilled before use. Acetonitrile, glacial acetic acid, chloroform, and methanol were purchased from Lab-scan Asia Co., Ltd. (Bangkok, Thailand). Ammonium acetate was obtained from Ajax Finechem (Seven Hills, NSW, Australia). Ammonium hydroxide solution (28% *w*/*w*) and anhydrous sodium sulfate were obtained from J.T.Baker^®^ (Radnor, PA, USA). The HPLC system was a Shimadzu LC-2030C 3D Plus Prominence-*i* Integrated HPLC System equipped with a photodiode array (Kyoto, Japan). 

### 4.3. Isolation of Mitragynine, Paynantheine, and Speciogynine

Kratom leaves were harvested from Nam Phu subdistrict, Ban Na San, Surat Thani, Thailand during June 2019. The crude methanol extract (CME) and the crude alkaloid extract (CAE) were prepared according to Nukitram et al. (2022) with a slight modification [36]. The isolation of mitragynine, paynantheine and speciogynine was performed using the methods described in [36,37]. The silica gel column was used to separate the CAE by eluting with a gradient of hexane: CHCl_3_, CHCl_3_: methanol, and methanol, afforded fraction 1, 2 and 3. Fraction 1 was fractionated over Sephadex LH-20 (2.5 × 60 cm), yielding mitragynine. Similarly, fraction 2 and 3 were purified using the chromatographic technique with *n*-hexane and ethyl acetate as eluent. The obtained fraction 2d was subsequently purified by a semi-preparative HPLC column (C18 VertiSep^TM^, 10 µm, 10 × 250 mm; isocratic elution with 20 mM ammonium acetate, pH 6.0: acetonitrile; 55:45), yielding paynantheine. Speciogynine was obtained from fraction 3. The summary of the isolation is shown in Appendix A. The structures of mitragynine, paynantheine and speciogynine were elucidated using ^1^H- and ^13^C-NMR. The purity was estimated by ^1^H-NMR and HPLC. The spectroscopic data were compared to those presented in a previous paper [30,37].

### 4.4. Macroscopic Examinations

Fresh kratom leaves were examined and photographed. The leaf blade, apex, base, and leaf margin were recorded, while the leaf size and weight were measured. 

### 4.5. Histolocalization Examination

Fresh kratom leaves were washed with distilled water. The transverse section of the fresh leaf, vein and petiole was stained with modified Dragendorff TS2 [38]. The alkaloid localization was observed using a light microscope.

### 4.6. Quantification of Alkaloid Contents 

Contents of mitragynine, paynantheine, and speciogynine were analyzed using HPLC analysis described previously with modifications [39].

#### 4.6.1. Sample Preparation

Kratom powder (50 mg) was accurately weighed and added to 1 mL of methanol. The extraction was performed with sonication in a water bath (maximum output) for 15 min, centrifuged (7000× *g* rpm, 10 min), and the supernatant was collected. The powder was exhaustively extracted by means of repeating the extraction five times. The supernatants were pooled and adjusted to volume as appropriate. The sample was filtered through a PTFE membrane (0.22 µm) before subjection to the HPLC system.

#### 4.6.2. Method Validation

The HPLC method was validated according to the ICH guidelines [40]. 

##### Specificity 

Peak purity was checked using a diode array detector. The HPLC chromatograms of the solvent, six independent kratom samples, and standard alkaloids were compared. 

##### Linearity and Range 

Mitragynine (MG), paynantheine (PAY), and speciogynine (SG) standards were individually prepared for working solutions, which were 1 mg/mL MG, 0.4 mg/mL PAY, and 0.5 mg/mL SG. The calibration curves were constructed ranging from 0.5 to 200 µg/mL for MG, 0.2 to 80 µg/mL for PAY, and 0.25 to 100 µg/mL for SG. The peak areas were plotted against the concentrations. A correlation coefficient (R^2^) value higher than 0.999 was required in the acceptance criteria. The limit of detection (LOD) and limit of quantification (LOQ) were determined from the signal to noise ratio, LOD (3:1 of S/N ratio) and LOQ (10:1 of S/N ratio), respectively.

##### Accuracy 

Working solutions of MG, PAY, and SG at three concentrations were spike into the 50 folds diluted sample. The accuracy was calculated using the following equation.
$$\mathrm{Recovery}\;\left(\%\right)=\frac{\left(\mathrm{Found}\;\mathrm{concentration}-\mathrm{Original}\;\mathrm{concentration}\right)\;}{\mathrm{Spiked}\;\mathrm{concentration}}\;\times \;100$$

##### Precision 

Intra-day precision was determined by injecting six replicated samples solution. For the inter-day precision, the triplicates of three concentrations were subjected to analysis for three consecutive days. The relative standard deviation (RSD) of the quantification was calculated. 

#### 4.6.3. HPLC Analysis

The HPLC analysis was performed using a Shimadzu LC-2030C 3D Plus Prominence-*i* Integrated HPLC system with a built-in autosampler, a column oven, a pump, and a diode array detector. The column used was the VertiSep^TM^ USP C18 HPLC column (4.6 × 250 mm, 5 µm) (Nonthaburi, Thailand). The column was isocratically eluted with 20 mM ammonium acetate, pH 6.0 and acetonitrile (35:65) at room temperature. The flow rate was set to 1 mL/min. The injection volume was 20 µL. The UV detector was set to 225 nm. The area under the curve was integrated. The amounts of MG, PAY, and SG were calculated by linear regression equation, and the values were expressed as the percent *w*/*w* of the dry weight. 

### 4.7. Soil Collection and Nutrient Analysis

Soil analysis was performed by the Agricultural Innovation and Management Division, Faculty of Natural Resources, Prince of Songkla University, Thailand. 

Topsoils were collected at a depth of 0–15 cm from the selected 20 locations of kratom cultivation across Thailand by the NCS, ONCB. One kilogram of soil was pooled from three areas surrounding the tree. The soil samples were air-dried at room temperature. Soil samples, passed through a 10 mesh sieve, were used to analyze the soil pH and electrical conductivity (immersed in water at a soil:water ration of 1:5 *w*/*v*). The organic matter (Walkley and Black method), total N (Kjeldahl method), available phosphorus (Bray II), extractable K, Ca, and Mg (1 M NH_4_OAc, pH 7.0), and extractable Fe, Mn, Zn, and Cu (DTPA extraction) were also investigated [41]. For soil physical analysis, soil texture was determined by means of the pipette method [42], and the soil water content at field capacity (0.1 bar) and permanent wilting point (15 bar) were determined by using pressure plate apparatus. 

### 4.8. Meteorological Data

Data on the average precipitation (mm) and temperature (°C) from June 2021 to May 2022 were retrieved from the Thai Meteorological Department, Thailand (https://www.tmd.go.th/; accessed on 24 November 2022). The request was made to collect the data from four weather stations, namely Petchabun, Bangkok (Don Muang), Ranong, and Surat Thani. The data showed an average amount for each month. 

### 4.9. Data Analysis 

The quantification of each sample was performed in triplicate. The results were shown as the mean ± standard deviation. The alkaloid contents of all samples were analyzed using one-way ANOVA (GraphPad), followed by Bonferroni’s multiple comparison test. *p*-values of 0.05 and 0.01 were considered significant for confidence intervals of 95%, and 99%, respectively. PAST 4.03 software was used to analyze the components of soil and to perform cluster analysis [43]. Principle component analysis was generated using the correlation method with variance of 34.6% for PC1 and 21.1% for PC2, and a biplot on the eigenvalue scale was generated. Hierarchical clustering of the samples was performed using Ward’s method. 

## 5. Conclusions

Based on the 745 samples collected from different seasons and locations in Thailand, we found that variations in the contents of pharmacologically active mitragynine, paynantheine, and speciogynine are dependent upon season and geographical origin. The tropical monsoon climate in the Malay Peninsula provides suitable weather and environmental conditions for kratom cultivation and producing significant amounts of alkaloids. The results obtained from this study will benefit Thai farmers in finding an optimal kratom plant and will facilitate further cultivation for medical purposes. 

## Figures and Tables

**Figure 1 plants-12-00949-f001:**
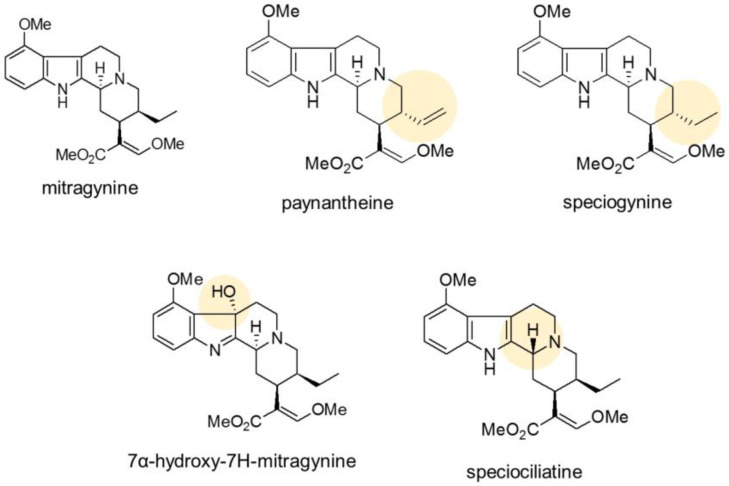
Chemical structures of alkaloids found in kratom leaves.

**Figure 2 plants-12-00949-f002:**
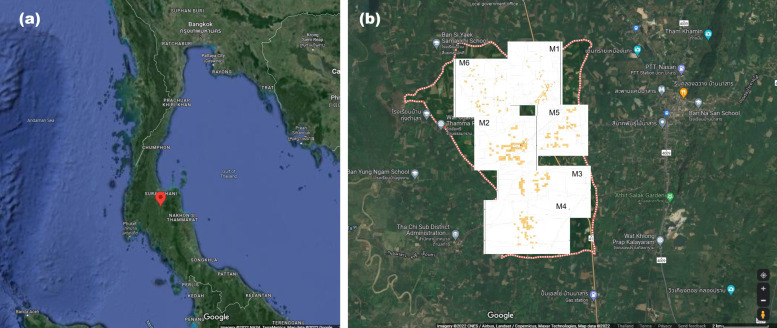
The mapping of kratom trees. (**a**) Location of Nam Phu subdistrict, Ban Na San, Surat Thani, Thailand, the first permitted area for growing kratom. (**b**) Mapping of kratoms tree in the area: red dots indicate the boundary of the subdistrict, the M1–M6 block indicate villages No. 1–6 and yellow dots indicate the location of kratom trees.

**Figure 3 plants-12-00949-f003:**
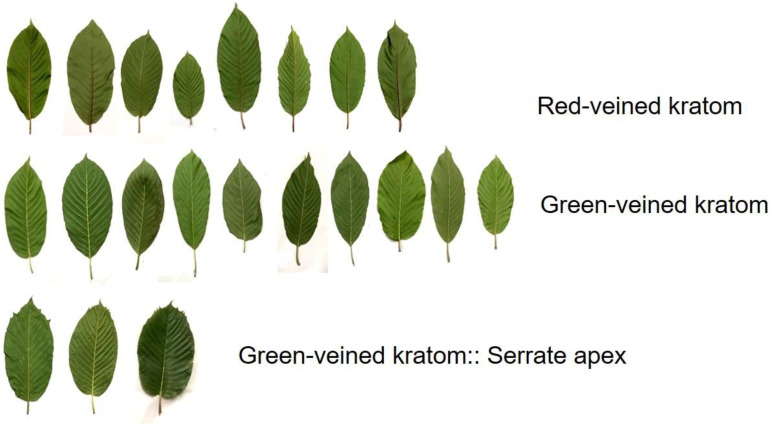
General appearances of kratom leaves collected from Nam Phu subdistrict, Ban Na San, Surat Thani during June 2019.

**Figure 4 plants-12-00949-f004:**
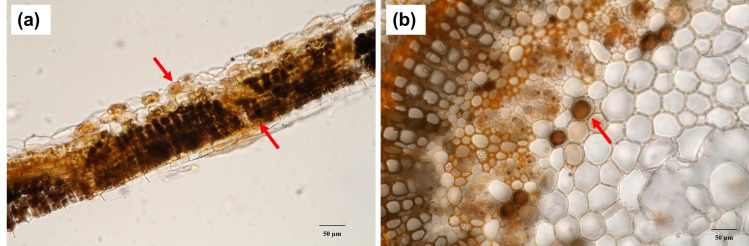
Histolocalization of alkaloid in leaf and petiole of *Mitragyna speciosa*. (**a**) Leaf palisade and spongy parenchyma containing alkaloid (arrow); (**b**) Idioblast containing alkaloid in petiole.

**Figure 5 plants-12-00949-f005:**
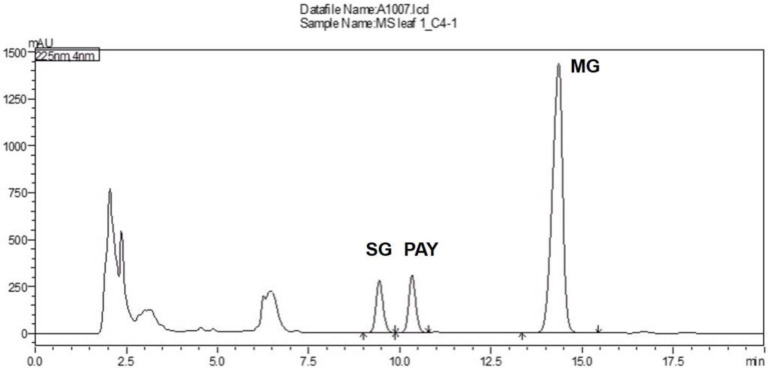
Typical HPLC chromatogram of the methanol extract of kratom leaves. SG: speciogynine; PAY: paynantheine; MG: mitragynine.

**Figure 6 plants-12-00949-f006:**
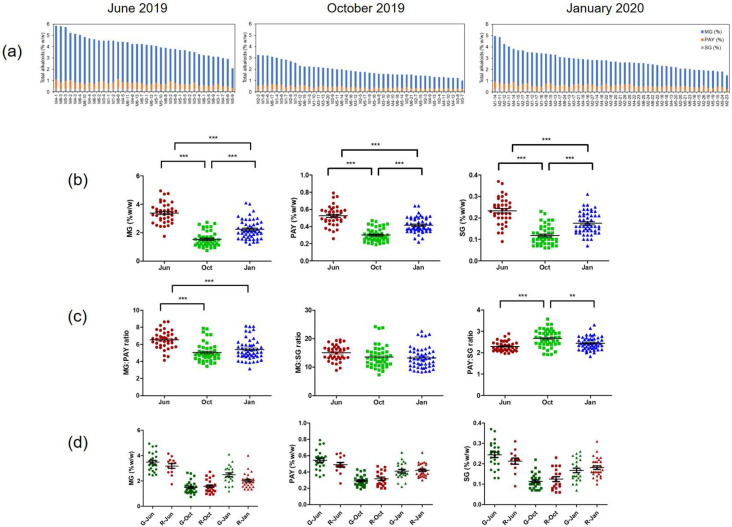
Distribution of alkaloids in kratom leaves, collected from Nam Phu subdistrict, Ban Na San, Surat Thani. (**a**) Total alkaloids in samples collected during June 2019, October 2019, and January 2020; (**b**) comparison of mitragynine (MG), paynantheine (PAY), and speciogynine (SG) contents in each collection; (**c**) plots of MG:PAY, MG:SG, and PAY:SG ratios in each collection; (**d**) multiple comparisons of MG, PAY, and SG contents in red- and green-veined kratom plants. The data were analyzed using one-way ANOVA, followed by Bonferroni’s multiple comparison test. **, *** demonstrate significant differences at *p*-value < 0.01, and 0.001, respectively.

**Figure 7 plants-12-00949-f007:**
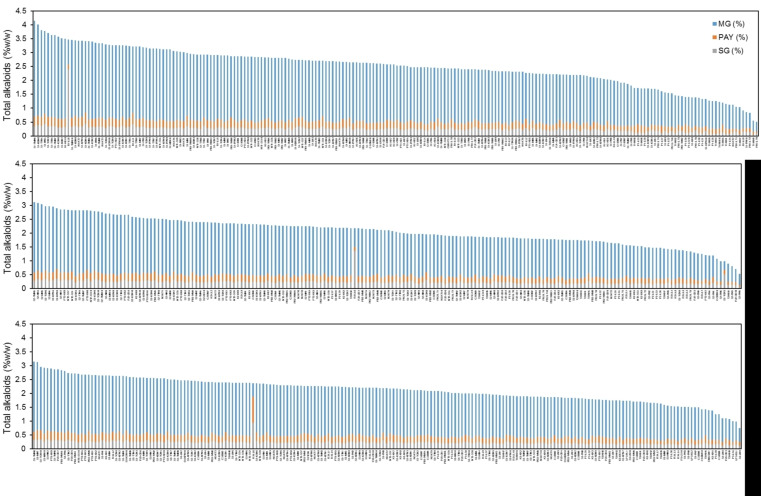
Total alkaloids in 611 samples, collected across Thailand in different collection periods. Red arrow indicates the high-SG-content kratom.

**Figure 8 plants-12-00949-f008:**
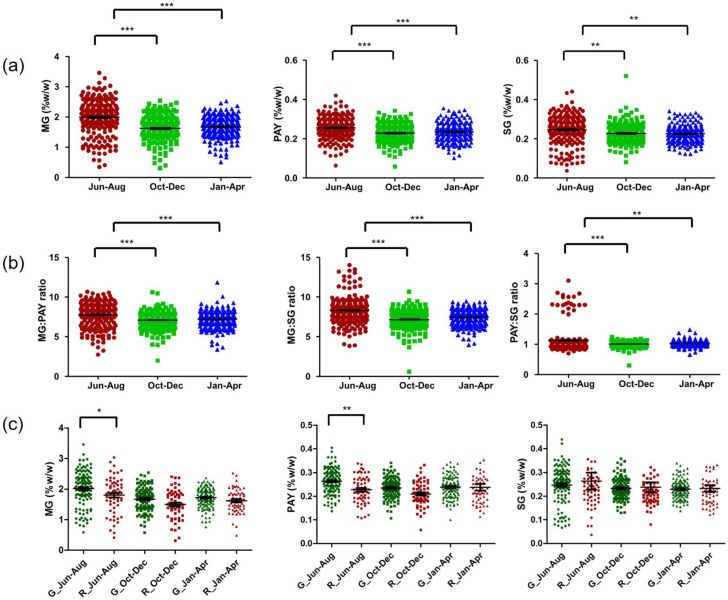
Distribution of alkaloids in kratom leaves across Thailand. (**a**) Comparison of MG, PAY, and SG contents in each collection; (**b**) plots of MG:PAY, MG:SG, and PAY:SG ratios; (**c**) multiple comparisons of MG, PAY, and SG contents in red- and green-veined kratom plants. The data were analyzed using one-way ANOVA, followed by Bonferroni’s multiple comparison test. *, **, *** demonstrate significant differences at *p*-value < 0.05, 0.01, and 0.001, respectively.

**Figure 9 plants-12-00949-f009:**
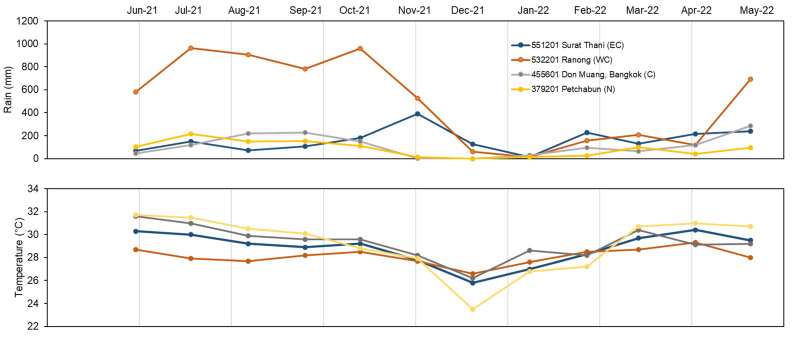
Annual precipitation and temperature during June 2021–May 2022. Data were kindly provided by Thai Meteorological Department, Thailand. (**a**) Rainfall in mm; (**b**) mean temperature in degrees Celsius. Colors indicate the weather stations, located in different parts of Thailand.

**Figure 10 plants-12-00949-f010:**
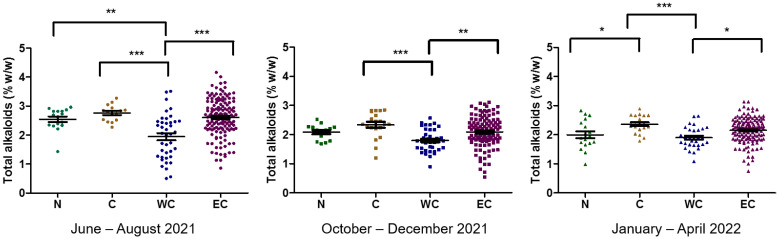
Distribution of alkaloids in different parts of Thailand collected during June 2021–April 2022. The data were analyzed using one-way ANOVA, followed by Bonferroni’s multiple comparison test. *, **, *** demonstrate significant differences at *p*-value < 0.05, 0.01, and 0.001, respectively.

**Figure 11 plants-12-00949-f011:**
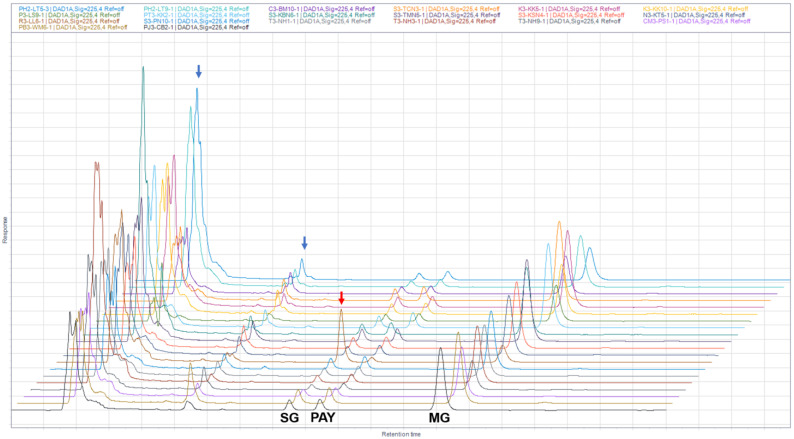
Alignment of HPLC chromatograms of 20 kratom samples. Sample codes are shown in the upper panel. SG, PAY, and MG stand for speciogynine, paynantheine, and mitragynine, respectively. Red arrow indicates the high SG content in the sample R3-LL6. Blue arrows indicate unidentified components in the extract.

**Figure 12 plants-12-00949-f012:**
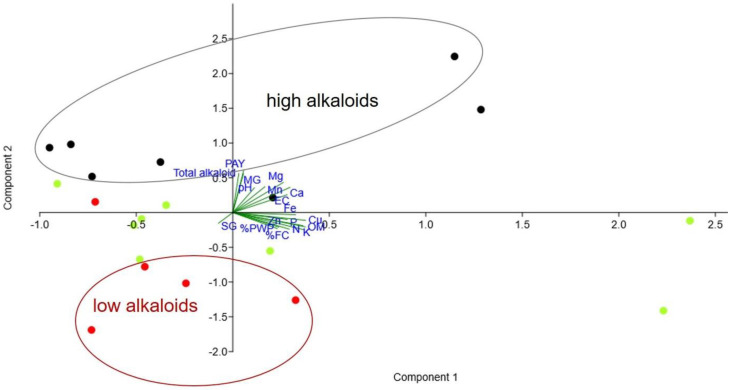
Principal component analysis of the correlation between soil nutrients and alkaloid contents using PAST 4.03 software. Diagram is plotted with biplot on eigenvalue scale. Black, green, and red dots indicate samples containing MG > 1.5%, 1–1.5%, and <1% *w*/*w*, respectively.

**Figure 13 plants-12-00949-f013:**
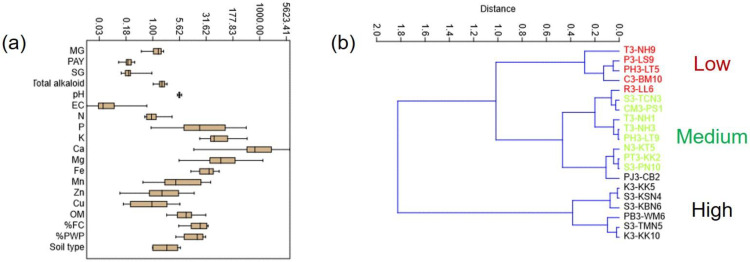
Statistical analysis of 20 kratom samples using PAST 4.03 software. (**a**) Whisker box plot correlation between soil nutrients (*x*-axis) and log value (*y*-axis), set at 95% interval with std. err. (**b**) Hierarchical clustering of total alkaloid contents (Ward’s method). Groups are highlighted: red: low-alkaloid group; green: medium-alkaloid group; black: high-alkaloid group.

**Figure 14 plants-12-00949-f014:**
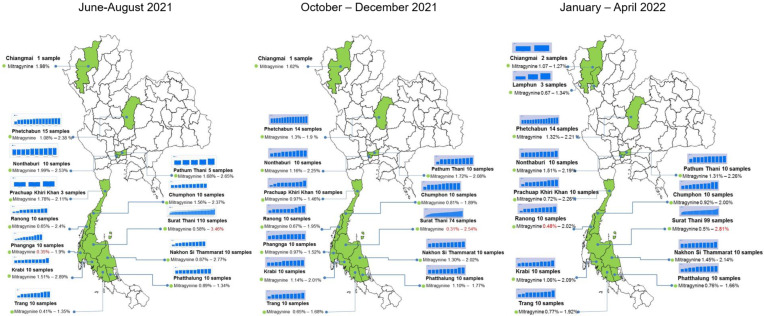
Mapping of MG contents in kratom grown in Thailand collected from June 2021 to April 2022. The name, location of the province, and the number of tested samples are stated. Values of the min--max of MG content are indicated. Red letters show min.-max. values in each season. Narcotic Crops Survey and Monitoring Institute, Office of the Narcotics Control Board, Thailand, kindly produced the pictures.

**Table 1 plants-12-00949-t001:** Validated parameters of the established HPLC method for alkaloid analyses.

Parameters ^1^	MG	PAY	SG
Linearity Range (µg/mL)	5–200	2–80	2.5–100
R^2^	0.9991	0.9990	0.9994
Precision (%RSD)			
Inter-day			
Low level; *n* = 3	2.939	6.321	5.826
Medium level; *n* = 6	2.592	3.227	3.628
High level; *n* = 3	3.331	3.132	3.446
Intra-day			
Medium level; *n* = 6	3.212	2.911	3.074
Accuracy (%Recovery)			
Low level; *n* = 3	94.27 ± 3.66	96.86 ± 3.90	94.61 ± 3.56
Medium level; *n* = 3	97.30 ± 1.12	104.54 ± 0.88	98.06 ± 0.97
High level; *n* = 3	96.48 ± 1.50	100.42 ± 1.80	98.04 ± 1.58
LOD (µg/mL)^2^	0.075	0.075	0.078
LOQ (µg/mL)^2^	0.2	0.2	0.2

^1^ The parameters were determined using a Shimadzu LC-2030C 3D Plus Prominence *i* Integrated HPLC system (Kyoto, Japan). ^2^ The values were visually evaluated from the S/N ratio.

**Table 2 plants-12-00949-t002:** Alkaloid contents, pH, electrical conductivity, and macronutrients in 20 samples collected from across Thailand.

Code	Sample Code	Alkaloid Contents (% *w*/*w*)				pH (1:5)	EC (dS/m)	Macronutrients				
		MG	PAY	SG	Total Alkaloid			N (g/kg)	P (mg/kg)	K (mg/kg)	Ca (mg/kg)	Mg (mg/kg)
1	S3-PN10	1.43	0.25	0.23	1.91	5.78	0.036	0.73	67.7	38	265.9	13.8
2	N3-KT5	1.48	0.19	0.18	1.85	6.01	0.023	0.68	75.4	33.9	1090.5	41.4
3	S3-KBN6	1.76	0.24	0.24	2.24	5.50	0.081	1.08	11.9	50.5	309.5	55.1
4	CM3-PS1	1.27	0.18	0.17	1.62	5.59	0.083	0.95	119.8	49.6	1418.1	45.0
5	C3-BM10	1.06	0.18	0.18	1.42	6.17	0.014	0.62	0.9	20.9	815.1	33.0
6	R3-LL6	0.48	0.11	0.94	1.53	5.37	0.027	0.74	3.7	52.0	14.2	5.5
7	PJ3-CB2	1.56	0.22	0.22	2.00	5.41	0.033	0.67	22.6	79.4	693.2	295.7
8	PB3-WM6	1.81	0.31	0.28	2.40	6.48	0.137	1.47	26.1	42.1	7079.5	1235.4
9	K3-KK5	1.74	0.20	0.20	2.14	5.55	0.030	0.60	6.1	42.6	2214.3	99.9
10	PH3-LT5	0.95	0.19	0.17	1.31	5.53	0.028	0.71	8.8	89.2	404.3	39.9
11	S3-TMN5	2.00	0.25	0.25	2.50	5.46	0.036	0.65	188.6	39.1	608.5	99.9
12	S3-TCN3	1.04	0.28	0.26	1.58	5.33	0.045	0.90	119.3	62.9	558.9	84.4
13	T3-NH1	1.31	0.20	0.18	1.69	5.11	0.031	1.09	3.9	139.4	157.9	24.7
14	K3-KK10	2.03	0.27	0.23	2.53	4.91	0.073	0.60	18.8	48.4	557.3	46.0
15	T3-NH9	0.77	0.13	0.13	1.03	5.50	0.047	1.14	37.2	81.3	2340.9	229.3
16	S3-KSN4	1.67	0.27	0.22	2.16	6.00	0.682	1.04	17.6	187.1	3848.9	617.6
17	P3-LS9	0.93	0.16	0.15	1.24	4.89	0.030	1.62	11.7	201.2	727.5	78.2
18	PT3-KK2	1.46	0.22	0.22	1.90	5.61	0.424	1.87	419.2	446.6	4562.1	609.6
19	T3-NH3	1.40	0.18	0.16	1.74	5.51	0.093	3.49	349.3	290.6	1958.3	88.3
20	PH3-LT9	1.43	0.17	0.15	1.75	5.73	0.048	1.28	7.1	53.7	752.2	131.8

**Table 3 plants-12-00949-t003:** Micronutrients, organic matters, soil type and water holding capacity in 20 samples.

Code	Sample Code	Micronutrients				OM (g/kg)	Soil Type	Water Holding Capacity	
		Fe (mg/kg)	Mn (mg/kg)	Zn (mg/kg)	Cu (mg/kg)			FC (%)	PWP (%)
1	S3-PN10	13.26	2.31	0.72	0.18	9.02	sandy clay loam	7.66	5.04
2	N3-KT5	49.22	3.40	7.40	1.87	4.77	clay loam	14.29	7.97
3	S3-KBN6	33.03	22.81	12.10	3.89	8.31	clay	25.87	19.09
4	CM3-PS1	20.34	3.97	4.07	1.00	9.62	sandy clay loam	11.34	4.98
5	C3-BM10	45.16	31.10	2.12	0.32	2.47	sandy clay loam	11.61	7.55
6	R3-LL6	15.16	0.53	0.45	0.15	10.08	clay	34.08	29.89
7	PJ3-CB2	50.47	42.37	1.16	0.90	3.39	clay loam	16.15	8.68
8	PB3-WM6	31.66	37.93	1.30	2.58	12.88	clay	30.73	26.15
9	K3-KK5	13.99	4.58	0.47	0.38	5.49	clay	18.35	13.16
10	PH3-LT5	44.02	0.56	0.13	0.41	5.32	clay	30.15	24.88
11	S3-TMN5	25.99	0.74	1.08	0.18	2.83	sandy clay loam	5.37	4.46
12	S3-TCN3	53.11	2.14	1.51	0.20	7.66	sandy clay	13.25	11.46
13	T3-NH1	11.77	0.58	0.12	0.21	5.71	silty clay	33.25	30.28
14	K3-KK10	21.81	4.76	3.09	0.34	2.93	sandy clay loam	7.41	5.28
15	T3-NH9	26.48	2.38	4.45	1.35	10.04	silty clay	17.16	16.43
16	S3-KSN4	63.42	41.32	1.07	1.34	16.16	clay	31.67	24.51
17	P3-LS9	44.72	21.82	3.76	3.73	13.60	clay	24.80	19.12
18	PT3-KK2	70.54	11.11	6.00	5.77	16.95	clay	34.76	25.77
19	T3-NH3	73.54	10.82	14.55	4.50	30.73	silty clay loam	35.03	21.40
20	PH3-LT9	42.92	4.27	5.60	2.28	10.39	silty clay	35.75	28.04

Remark: OM: organic matter; FC: field capacity; PWP: permanent wilting point.

## Data Availability

Not applicable.

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
