# Peer review of "Seasonal and Geographic Variation in Alkaloid Content of Kratom (*Mitragyna speciosa* (Korth.) Havil.) from Thailand"

_plants, 2023, doi:10.3390/plants12040949_

Round 1

Reviewer 1 Report

The authors presented data on the distribution of alkaloids in kratom grown in Thailand. The two stages of collections covered the regions of southern, central and northern Thailand and different seasons. The influence of various factors (such as soil nutrients and climate) on alkaloid content was analysed. The macroscopic and microscopic structure of kratom's leaf are also illustrated.

The paper is well written and gives a good overview of the issue.  The finding presentation is very systematic, detailed, and transparent. However, there are errors in the numbering of pages, tables, figures (I couldn't find: Table A2, Figure A3, ...). I  recommend the manuscript publication in “Plants” journal after these minor errors have been corrected.

Author Response

The paper is well written and gives a good overview of the issue.  The finding presentation is very systematic, detailed, and transparent. However, there are errors in the numbering of pages, tables, figures (I couldn't find: Table A2, Figure A3, ...). I  recommend the manuscript publication in “Plants” journal after these minor errors have been corrected.

Ans. Thank you very much. There were typo errors; it should be Table S2, Figure S3 (highlighted in yellow on p. 4, 6). There is some mistake in the numbering of pages, and I could not change it. I will consult the journal to help with this.

Reviewer 2 Report

The English is not the best but fixable, and there is a fair bit of unnecessary repetition of text. Not really the most interesting of papers but given the socioeconomic significance of kratom I can recommend publication of this rather straightforward work.

On page 6 should be Figure S3, not A3. I could not find Table "A3".

Please check the collection dates, I found them confusing and there may be inconcsistencies.

Author Response

The English is not the best but fixable, and there is a fair bit of unnecessary repetition of text. Not really, the most interesting of papers but given the socioeconomic significance of kratom I can recommend publication of this rather straightforward work.

Ans. The manuscript is submitted for English improvement.

On the page, six should be Figure S3, not A3. I could not find Table "A3".

Ans. There were typo errors; it should be Table S2, Figure S3 (highlighted in yellow on p. 4, 6).

Please check the collection dates, I found them confusing and there may be inconsistencies.

Ans. Thank you very much. Especially in set 2, we edited and revised the collection date accordingly on pages 6 and 21, Figure S1, and the abstract. All are highlighted in yellow.

Reviewer 3 Report

This article deals with the distribution of 3 alkaloids (mitragynine (MG), paynan- 20 theine (PAY) and speciogynine (SG)) in the leaves of the kratum plant collected in different parts of Thailand and at different times of the year. Three types of experiments can be distinguished. In the first part of the work, alkaloids were isolated from kratom leaves collected in Nam Phu sub-district in three different growing seasons. The HPLC method for determining the concentration was then validated and the authors reported the morphological and macroscopic characteristics of the leaves of the plant. In the second part of the experiment, kratom leaves were collected from 19 areas in Thailand in three seasonal periods and the concentration of alkaloids in them was determined. . Metrological data were also given to explain the different concentrations of the alkaloids. Finally, the soil in 20 areas where the leaves were collected was analysed and the results were presented relating the amount of alkaloids to the type of soil.

The manuscript contains many interesting results that can be applied to the selection of conditions for the cultivation of kratom, especially in Thailand. However, the manuscript is written very confusingly in some parts, making it difficult to follow the concept of the research and the results. Therefore, I suggest some changes that could contribute to the quality of the manuscript.

Title: I think the title should be changed and the geographical origin should be added to the title of the paper
Summary: At the end of the summary, add the results obtained based on the climate and soil analysis.

Introduction: After line 48, transfer part of the text from lines 63-68. Then comes the part mentioning the use of kratom for medicinal purposes (lines 49-52), then the chemical composition (lines 68-80) and then lines 52-58.

Results

Lines 179-180 What are Figure and Table A3

Subtitle 2.5 Effect of climate and geographical origin on alkaloid content - rewrite the subtitle as only meterological data is given and the effect of climate on alkaloid composition is not shown as it was for geographical origin.

Transfer Lines 328-336 to the beginning of part 2.6 Effect of soil nutrients on alkaloids contents and delete Lines 298-304

Discussion

Meanwhile, several researchers around the world reported the pharmacological activities and plausible mechanism in in vitro and in vivo models - add references.

Author Response

This article deals with the distribution of 3 alkaloids (mitragynine (MG), paynan- 20 theine (PAY) and speciogynine (SG)) in the leaves of the kratum plant collected in different parts of Thailand and at different times of the year. Three types of experiments can be distinguished. In the first part of the work, alkaloids were isolated from kratom leaves collected in Nam Phu sub-district in three different growing seasons. The HPLC method for determining the concentration was then validated and the authors reported the morphological and macroscopic characteristics of the leaves of the plant. In the second part of the experiment, kratom leaves were collected from 19 areas in Thailand in three seasonal periods and the concentration of alkaloids in them was determined. . Metrological data were also given to explain the different concentrations of the alkaloids. Finally, the soil in 20 areas where the leaves were collected was analysed and the results were presented relating the amount of alkaloids to the type of soil.

The manuscript contains many interesting results that can be applied to the selection of conditions for the cultivation of kratom, especially in Thailand. However, the manuscript is written very confusingly in some parts, making it difficult to follow the concept of the research and the results. Therefore, I suggest some changes that could contribute to the quality of the manuscript.

Title: I think the title should be changed and the geographical origin should be added to the title of the paper
Summary: At the end of the summary, add the results obtained based on the climate and soil analysis.

Ans. We changed the title as suggested to ‘Seasonal and Geographic Variation on Alkaloids Distribution in Kratom [Mitragyna speciosa (Korth.)Havil.] Grown in Thailand’.

In the abstract: we revised by adding the results based on meteorological data and soil analysis.

Introduction: After line 48, transfer part of the text from lines 63-68. Then comes the part mentioning the use of kratom for medicinal purposes (lines 49-52), then the chemical composition (lines 68-80) and then lines 52-58.

Ans. We edited as suggested. Thank you very much.

Results

Lines 179-180 What are Figure and Table A3

Ans. There are typo errors. It is Figure S3.

Subtitle 2.5 Effect of climate and geographical origin on alkaloid content - rewrite the subtitle as only meterological data is given and the effect of climate on alkaloid composition is not shown as it was for geographical origin.

Ans. The title is changed to ‘Meteorological and geographic data’.

Transfer Lines 328-336 to the beginning of part 2.6 Effect of soil nutrients on alkaloids contents and delete Lines 298-304

Ans. We edited it as suggested. Thank you very much.

Discussion

Meanwhile, several researchers around the world reported the pharmacological activities and plausible mechanism in in vitro and in vivo models - add references.

Ans. The references are added.

Reviewer 4 Report

·    Interested work, great effort, and minor corrections are needed!

There are some comments

1. The abstract is confusing, especially the methods part I think the author could rewrite that part [L:21-27] as written in the study aims [L:100-102] it’s more clear and I recommend that the author followed the style of a structured abstract.

2. Grammar check is needed for the whole article!  

3. Typo mistakes for exp. L: 44, 49,53,58, 84 it seems there is no space between the words, etc.

4. It’s preferred to move Figure 1 after the text!

5. L: 116 and L: 119 Do you mean Table S1 and Table S2 instead of Table A1 and A2? So please change them to be the same as you mentioned in the supplementary file. Same for Figure A3 and Table A3, etc.

6. L: 179 where is Table A3 or do you mean Table 2?

7. Results section: Figures 6 and 11 are missing to mention them inside the text!

8. L:203-204 The sequence of the periods is not the same as in the figure rewrite the sentence, please!  

Author Response

Interested work, great effort, and minor corrections are needed!

There are some comments

  1. The abstract is confusing, especially the methods part I think the author could rewrite that part [L:21-27] as written in the study aims [L:100-102] it’s more clear and I recommend that the author followed the style of a structured abstract.

Ans. We edited it as suggested. Thank you very much.

  1. Grammar check is needed for the whole article!

Ans. The manuscript is submitted for English improvement.

  1. Typo mistakes for exp. L: 44, 49,53,58, 84 it seems there is no space between the words, etc.

Ans. We edited it as suggested. Thank you very much.

  1. It’s preferred to move Figure 1 after the text!

Ans. We edited it as suggested. Thank you very much.

  1. L: 116 and L: 119 Do you mean Table S1 and Table S2 instead of Table A1 and A2? So please change them to be the same as you mentioned in the supplementary file. Same for Figure A3 and Table A3, etc.

Ans. There were typo errors; it should be Table S2, Figure S3 (highlighted in yellow on p. 4, 6).

  1. L: 179 where is Table A3 or do you mean Table 2?

Ans. It was a mistake. It should be Figure S3, not Table.

  1. Results section: Figures 6 and 11 are missing to mention them inside the text!

Ans. We edited it as suggested. Thank you very much.

  1. L:203-204 The sequence of the periods is not the same as in the figure rewrite the sentence, please!

Ans. The sequence is corrected, as samples collected in October 2019 accumulated the lowest amount of alkaloids.
